# Optimizing Antibiotic Treatment Strategies for Neonates and Children: Does Implementing Extended or Prolonged Infusion Provide any Advantage?

**DOI:** 10.3390/antibiotics9060329

**Published:** 2020-06-17

**Authors:** Paola Costenaro, Chiara Minotti, Elena Cuppini, Elisa Barbieri, Carlo Giaquinto, Daniele Donà

**Affiliations:** 1Division of Paediatric Infectious Diseases, Department for Women’s and Children’s Health, University of Padova, 35128 Padova, Italy; elisa.barbieri.5@phd.unipd.it (E.B.); carlo.giaquinto@unipd.it (C.G.); daniele.dona@phd.unipd.it (D.D.); 2Department for Women’s and Children’s Health, University of Padova, 35128 Padova, Italy; minotti.chiara@gmail.com (C.M.); elenacuppini28@gmail.com (E.C.); 3Paediatric Network for Treatment of AIDS (Penta) Foundation, 35128 Padua, Italy

**Keywords:** antibiotic, time-dependent, continuous infusion, children, pediatric

## Abstract

Optimizing the use of antibiotics has become mandatory, particularly for the pediatric population where limited options are currently available. Selecting the dosing strategy may improve overall outcomes and limit the further development of antimicrobial resistance. Time-dependent antibiotics optimize their free concentration above the minimal inhibitory concentration (MIC) when administered by continuous infusion, however evidences from literature are still insufficient to recommend its widespread adoption. The aim of this review is to assess the state-of-the-art of intermittent versus prolonged intravenous administration of antibiotics in children and neonates with bacterial infections. We identified and reviewed relevant literature by searching PubMed, from 1 January 1 2000 to 15 April 2020. We included studies comparing intermittent versus prolonged/continuous antibiotic infusion, among the pediatric population. Nine relevant articles were selected, including RCTs, prospective and retrospective studies focusing on different infusion strategies of vancomycin, piperacillin/tazobactam, ceftazidime, cefepime and meropenem in the pediatric population. Prolonged and continuous infusions of antibiotics showed a greater probability of target attainment as compared to intermittent infusion regimens, with generally good clinical outcomes and safety profiles, however its impact in terms on efficacy, feasibility and toxicity is still open, with few studies led on children and adult data not being fully extendable.

## 1. Introduction

Infections acquired during hospitalization (HAIs) are particularly challenging among vulnerable populations, particularly neonates and children affected by chronic conditions such as immune deficiency or when admitted to the intensive care unit. The emergency of multidrug resistant bacteria as a common cause of HAIs, with very limited treatment options available for pediatric use, increases the need of optimizing the use of currently employed antibiotics, focusing on the best dosing strategy to improve overall outcomes as well as to limit the further development of antimicrobial resistance. 

Prolonged and continuous infusion of time-dependent antibiotics aimed at increasing the probability of attaining pharmacokinetic/pharmacodynamic (PK/PD) targets may be considered to address challenges related to difficult-to-treat pathogens and improve efficacy [1]. Although conflicting results from studies conducted on adults cannot allow the recommendation of a widespread adoption of continuous infusion antibiotics in place of intermittent infusions [2,3,4], recent evidence suggest that administering beta-lactam antibiotics by prolonged or continuous infusions may reduce mortality, particularly in the case of difficult-to-treat pathogens such as *Pseudomonas aeruginosa* and/or bacteria with high minimal inhibitory concentration (MIC) [5,6,7,8]. 

Several aspects have to be taken into account in selecting the most appropriate antimicrobial treatment for patients with suspected or confirmed infection. Several aspects have to be considered in the decisional approach, including pathogens’ characteristics and their sensitivity/resistance pattern to common antibiotics, the drugs’ intrinsic pharmacodynamic features, the infection site and the patient’s pathophysiology [9]. Composing the antimicrobial treatment puzzle is even more complex in neonates and children, due either to their age-related changes in physiological characteristics and to the limited antibiotic options that have already been approved for the pediatric population. Indeed, age-related differences in physiological characteristics have to be taken into account as they have an impact on the pharmacokinetic parameters of the administered drug, particularly due to variations in absorption, distribution, metabolism and excretion [10]. Moreover, further variations in pharmacokinetics of mainly hydrophilic antimicrobials occur in critically ill neonates and children [11,12]. 

Among all antibiotics, it has been shown that the bactericidal activity of time-dependent antibiotics, such as beta-lactams and oxazolidinones, is related to the duration of the maintenance of their free concentration above the minimal inhibitory concentration (MIC) during each dosing interval [13] while the killing activity of vancomycin is related to the area under the curve (AUC) and MIC ratio (AUC/MIC) [14]. Standard efficacy of beta-lactams is ensured when the duration of time the concentration exceeds the MIC (t > MIC) is at least 50% of the dosage interval, whereas a t > MIC of 100% of the dosage interval should be ensured to reach optimal exposure, particularly in immune-compromised patients [12,15,16,17]. Indeed, a further improvement in efficacy of time-dependent antimicrobials has been observed for plasma concentrations four to five folds greater than the MIC [12,18]. In the case of vancomycin, it has been shown that standard efficacy is ensured reaching an AUC/MIC ratio higher than 400 [14]. Based on these pharmacodynamic determinants, prolonged or continuous infusion of time-dependent antimicrobials may represent the best form of administration to manage severe infections/sepsis by ensuring the highest steady-state concentrations. 

In 2012, Walker and colleagues examined intermittent versus continuous administration of such antibiotics in children [19], reporting a lack of evidence in this particular population. So far, despite an observed increasing emergency of multidrug resistant bacteria even among the pediatric population, few antimicrobials have been in the pipeline for pediatric use, confirming the need of review the most appropriate use of commonly used drugs aimed at preserving as much as possible their efficacy and improve patients’ outcome.

The aim of this narrative review is to evaluate the state-of-the-art in the literature on the intravenous use of systemic antibiotic treatment for both children and neonates with infections due to either Gram-positive and Gram-negative bacteria, particularly focusing on intermittent versus prolonged infusion.

## 2. Methods

### 2.1. Search Strategy

We identified and reviewed relevant literature by searching PubMed, from 1 January 2000 to 15 April 2020. Within the research strategy used for PubMed, combinations of the following search terms were used: “antibiotic”, “antimicrobials”, “children”, “paediatric”, “pediatric”, “neonate”, “infusion”, “extended infusion,” “prolonged infusion,” “continuous infusion,” “continuous administration,” “dosing regimen,” “dosing regimens,” “continuous vs. intermittent,” “extended vs. intermittent,” “standard vs. prolonged,” “extended vs. standard,” or “intermittent vs. prolonged,” “piperacillin/tazobactam”, “meropenem”, ceftazidime”, “cefepime”, “vancomycin”, with a filter of “text availability: abstract, free full text and full text”. Moreover, a reference list from eligible articles was reviewed to identify other potentially relevant studies. The last search was conducted on 15 April 2020. 

### 2.2. Selection Criteria

In the current review, we included studies comparing the outcomes of different infusion regimens in the pediatric population, focusing on intermittent versus prolonged/continuous infusion rates. Randomized controlled trials, pharmacokinetic/pharmacodynamic studies, observational studies, and case series were included if involving pediatric patients (age 0–18 years) and if comparing prolonged or extended versus intermittent infusion of time-dependent antibiotics. 

Manuscript comments, letters, editorials, conference abstracts and opinion articles were excluded. 

### 2.3. Data Collection 

Data were extracted using a standardized data collection form, which summarized information about authors, year of publication, study design, country, study period, setting, multicentric involvement, type of intervention, and main results.

## 3. Results and Discussion

Of 38,906 titles and abstracts, 114 were eligible for inclusion in this review, and ten studies were included, nine published in English and one in Spanish. 

### 3.1. Glycopeptides

This class of antibiotics exhibits time-dependent bactericidal activity against most gram-positive bacteria, including methicillin-resistant *Staphylococcus aureus* (MRSA) and *Enterococcus* species; oral administration of glycopeptides is also recommended for the management of *Clostridium difficile* infection [20].

### 3.2. Vancomycin

Vancomycin inhibits cell wall synthesis by binding to the d-Ala-d-Ala terminal of the growing peptide chain during Gram positive cell wall synthesis. Details about spectrum of activity, adverse effects and hints to antibiotic resistance can be found in Table 2. Its volume of distribution is 0.4–1 L/kg [14] and protein binding is thought to be approximately 50%, with estimated variability [21]. Vancomycin is usually administered intravenously, over at least 1 h. In children aged from 1 month to 12 years with normal renal function the advised intravenous daily dose is 40–60 mg/kg, administered in four divided doses [22] while in neonates up to 1 month, the recommended initial dose is 15 mg/kg followed by a maintenance dose of 10 mg/kg every 12 h in the first week of life and then once every 8 h until the age of one month. A wide inter-individual variability has been shown in preterm and term neonates [23].

Serum levels of vancomycin should be monitored and a target trough concentration goal of 15–20 μg/mL is recommended for serious, complicated infections, including methicillin-resistant staphylococcal-related infections [24,25].

### 3.3. Pharmacokinetic/Pharmacodynamic Data 

The most useful target parameters to evaluate vancomycin PK/PD correlation are the area under the curve (AUC) and MIC. According to Rybak et al., an AUC/MIC ratio higher than 400 is related to a plasma trough level above 15 μg/mL, assuming 1 mg/L MIC or less [14].

In the pediatric population, PK/PD data on vancomycin are limited. A model study reports that the current empiric recommended vancomycin dose in children of 40 mg/kg/day is unlikely to achieve the recommended pharmacodynamic target of AUC 24/MIC > 400 in case of methicillin-resistant *S. aureus* (MRSA) with MIC of 1.0 μg/mL or greater, suggesting that dose should be increased to 60/mg/kg/die [24]. Another prospective study reports that to achieve more rapidly the PK/PD targets in burn children with normal renal function, an initial dose of approximately 90–100 mg/kg/day should be recommended [26]. Similar findings were reported by other studies based on PK data from children admitted to intensive care units (PICU) and Monte Carlo simulations, suggesting that on the basis of age, serum creatinine, and MIC distribution, a higher dose of vancomycin (60 to 70 mg/kg/day) could be necessary to achieve AUC/MIC ≥ 400 [27,28,29]. In conclusion, most studies agree on the fact that PK models and Bayesian approaches may help in improving individualized target attainment [30].

### 3.4. Clinical Outcome

Although evidence from adults showed that continuous infusion of vancomycin (CIV) decreases the risk of nephrotoxicity and the incidence of infusion-related reactions while also decreasing time to therapeutic concentrations and drug cost at the same time, compared to IIV [31,32,33], limited evidence is currently available for the pediatric population. 

Zylbersztajn et al. published a case series of six children between two months and seven years of age, being initially treated for MRSA sepsis with vancomycin 40 and 60 mg/kg/day every 8–6 h. Because of poor outcome, they were all shifted to continuous infusion at 50 mg/kg/day, for 9 to 18 days. They reached blood levels between 10 and 25 ug/mL, with a favorable outcome and negativization of cultures, with no signs of nephrotoxicity [34].

A randomized trial conducted by Gwee et al. compared the treatment outcome of continuous infusion of vancomycin (20–50 mg/kg/day) with the intermittent administration of 15 mg/kg/dose every 24, 12, 8 or 6 h according to gestational age in 111 infants admitted in NICU and PICU with suspected sepsis. Both administration regimens were related to clinical improvement and no significant side-effects were reported. However, only 41% (21 out of 51) of children of the intermittent intravenous (IIV) group achieved a target concentration of 15 to 25 mg/L at first steady-state level, compared to the 85% (45/53) of continuous intravenous (CIV) group [35]. 

The same outcome was evaluated in a retrospective study of 77 preterm infants (gestational age < 34 weeks) with suspected late-onset sepsis (of those, an MRSA infection was diagnosed in 19 patients), treated with 20 mg/kg/day of vancomycin: 48 h after treatment initiation, 52.8% of infants of CIV achieved therapeutic levels, compared to 34.1% of patients of the IIV group. Microbiological outcomes and clinical responses did not differ significantly between the two groups [36].

Hurst and colleagues retrospectively evaluated the achievement of serum vancomycin goal concentration at different age range and the associated safety and tolerability among a cohort of 240 pediatric patients who were shifted from IIV to CIV. In their cohort, the average final total daily dose of CIV required to attain a therapeutic serum vancomycin concentrations (SVCs) was 54% to 64% of the final IIV dosing, suggesting that CIV has the potential to result in goal SVCs with much lower dosing than IIV [37]. As for safety outcomes, 19 patients of CIV had a mild to moderate decrease in creatinine clearance while a 17-year-old patient with a goal of 15 to 20 μg/mL had renal injury and one more 10-year-old patient with SVC 10 to 15 μg/mL presented renal failure, both being on other nephrotoxic medications, with serum creatinine level being back to normal at discharge. 

As for compliance outcomes and availability for outpatients, there are to date no available data in the pediatric population, but evidence from studies involving adults seem to confirm a good tolerance, efficacy and safety profile for CIV with elastomeric pumps in home settings [38,39]. 

### 3.5. B-Lactams Antibiotics 

Beta lactams are a family of time-dependent pharmacodynamic antibiotics that include penicillins, cephalosporins, carbapenems and monobactams, acting as inhibitors of bacterial cell wall synthesis [40]. Several studies showed that the bactericidal activity of beta lactams is highly predicted by the time during which the non–protein bounding drug concentration exceeds the MIC (fT > MIC) of the organism, at the site of the infections [41]. Although the precise fT > MIC varies for different drug-bacteria combinations, a near-maximal bactericidal effect has been reported when the free drug concentration exceeds the MIC for 60–70%, 50%, and 40% of the dosing interval for the cephalosporins, penicillins, and carbapenems, respectively [42,43]. Furthermore, it has been demonstrated that the bactericidal activity is maximized if free drug concentration remains four times higher than the MIC of the bacteria, without any advantage in further increasing the dose [15]. Prolonging the time of drug administration through either extended or continuous infusion may lead to maintain a stable trough concentration, minimizing high peak concentrations and therefore achieving such pharmacodynamic target more successfully than intermittent bolus dosing. Although some findings from RCTs conducted among the adult population provided conflicting conclusions [3,4], recent evidence has been supporting the use of prolonged or continuous infusion to reduce mortality and improve clinical outcomes [5,6,7,8]. 

### 3.6. Piperacillin/Tazobactam

Piperacillin/tazobactam (TZP) is a broad spectrum beta-lactam/beta-lactamase inhibitor combination widely used in hospitalized children with either suspected or documented severe infections, due to its wide spectrum of activity against Gram-positive/Gram-negative aerobic and anaerobic pathogens [44,45,46,47,48], including *Pseudomonas aeruginosa* (details in Table 2). Few pharmacokinetic data are available for piperacillin alone or piperacillin-tazobactam in children less than the age of two years, thus limiting its use in this population. 

### 3.7. Pharmacokinetic/Pharmacodynamic Data

This drug is routinely administered with a 30-min or 1-h infusion time, as recommended for any other beta-lactams. However, recent studies challenge currently used dosing regimens and way of administration. According to De Cock et al., a Monte Carlo simulation conducted in a population of critically ill children receiving 75–100 mg/kg piperacillin every 6–8 h as a short infusion lead to very low probability of target attainment (PTA), estimated between 5.9% and 34% for piperacillin, potentially leading to subtherapeutic treatment. On the other side, continuous or prolonged (every 4 h) infusions met the PTA criterion for piperacillin [49], defined as obtaining a PTA value of ≥ 90%, as previously established [50]. Similar findings were reported by other studies, based on pK data from children admitted to intensive care unit (PICU), suggesting that prolonged or continuous administration may be more convenient, particularly in case of infection due to Gram-negative bacteria with higher MICs [51,52]. The same conclusions were highlighted considering as referral population either neonates and infants admitted to neonatal intensive care unit (NICU) [53] or febrile children with cancer [54,55], particularly when maximizing the dose of TZP (e.g., 400 mg/kg/day) [56]. Lastly, according to Thibault et al., infants and children older than six months seem to need extended TZP administrations to achieve a favorable PD target. The proposed dosage against bacteria with MICs of up to 16 mg/L, in patients from six months to six years of age, was 130 mg/kg/dose every 8 h infused over 4 h (total daily dose, 390 mg/kg/day; total infusion time, 12 h) [57].

### 3.8. Clinical Outcome

To date, few pieces of evidence evaluate the clinical impact of prolonged or continuous administration of piperacillin/tazobactam in children. Only one non-blinded RCT, published in 2019 by Solórzano-Santos et al., evaluated the clinical efficacy of TZP administered through continuous rather than intermittent infusion in onco-hematological children with febrile neutropenia. Continuous infusion included a loading dose of 75 mg/kg over 30 min, followed by a 24 h continuous infusion of 300 mg/kg/day. Among 176 episodes assessed, no statistically significant difference was found in fever resolution, clinical cure rate or mortality between the continuous Infusion group and the intermittent administration group, with respectively 16 and 13 cases of treatment failure; one patient in each group died [58]. 

A retrospective case series published in 2017 showed that 29 (74%) out of 39 children of five years (IQR 2–9) median age affected by an Enterobacteriaceae (mostly *E. coli* and *K. pneumoniae*) invasive infection and treated with prolonged TZP infusion achieved clinical cure at 21 days after treatment initiation. Although 38.5% (*n* = 15) of patients had a readmission after 30 days, no deaths were reported in this cohort. Indeed, adverse effects related to extended infusion TZP were not experienced [59].

### 3.9. Ceftazidime

Ceftazidime is a third-generation cephalosporin frequently used in pediatric patients, being active against several Gram-positive and Gram-negative germs, including *P. aeruginosa,* and having a more favorable safety profile, if compared to other cephalosporins. Details can be found in Table 2. This drug is commonly prescribed in neonates with body weight > 1 kg and with less than seven days of life at 50 mg/kg/dose every 12 h while increasing doses are suggested if post-natal age is 8 to 28 days [60], up to 150 mg/kg/day divided every 8 h in case of meningitis [61]. 

According to the Red Book, the treatment of mild to moderate infections in infants, children and adolescents requires 90 to 250 mg/kg/day divided every 8 h (maximum daily dose 3 g/day), while severe infections 200 mg/kg/day divided every 8 h (maximum daily dose: 6 g/day, with even higher doses (300 mg/kg/day) being recommended for cystic fibrosis patients) [60].

### 3.10. Pharmacokinetic/Pharmacodynamic Data

Ceftazidime has an almost complete renal excretion (80–90%) and has a low protein binding profile (10%) [62]. It is usually administered in children with intermittent intravenous infusions over 15 to 30 min, however the use of prolonged or continuous administration has been taken into consideration among the pediatric population, in order to optimize the time of free plasma drug concentrations above the MIC, thus enhancing its time-dependent antibacterial activity. Findings reported by several pharmacokinetics/pharmacodynamics studies suggested that continuous infusion administration of ceftazidime could optimize the time above the MIC for the pathogen therefore allowing higher concentrations to be achieved in tissues [63]. More recently, Cojutti et al. led a population pharmacokinetic analysis of continuous-infusion ceftazidime hematopoietic stem cell transplantation (HST). Ceftazidime steady-state (Css) plasma concentrations were monitored and among 46 children with 70 ceftazidime Css values considered, at the EUCAST clinical breakpoint of 8 mg/L for *P. aeruginosa*, Monte Carlo simulations showed that continuous-infusion ceftazidime doses of 4–6 g/day attained optimal PTAs (>90%) across most of 16 different clinical scenarios based on four classes of eGFR and body surface area [64]. Dalle et al. confirmed the feasibility and safety of the continuous infusion regimen, as reported in their study on the pharmacokinetics of ceftazidime based on 20 febrile neutropenic pediatric patients [65]. 

### 3.11. Clinical Outcome

Few studies evaluated the clinical impact of continuous rather than intermittent administration of ceftazidime in the pediatric population. 

In their prospective study, Rappaz et al. compared the treatment outcome of continuous and intermittent infusion of ceftazidime, both administered in different times at the same cohort of 14 children affected by cystic fibrosis. In the intermittent administration regimen group, the mean drug trough level concentration in serum was highly variable and 32% of samples had values below the MIC of pathogen isolated in sputum (*P. aeruginosa*) while the continuous infusion regimen ensured higher serum ceftazidime levels with no values below the MIC [66]. Despite this, both regimens showed a clinical improvement in terms of several pulmonary, inflammatory and nutritional variables, assessed at first and last day of treatment [66]. In addition, no significant side-effects were reported. 

Lastly, according to the randomized crossover study of Hubert et al. conducted among children affected by cystic fibrosis, ceftazidime administered in a continuous infusion was as efficient as short infusion regimens and quality-of-life scores were comparable for the two groups. There were no reported toxicity issues [67].

As for continuous infusion ceftazidime for the treatment of outpatients, Jones et al. demonstrated its safe and effective use with once-daily changes of infusion device, provided the concentration and temperature of the infusion solution is controlled, in order to limit its degradation and pyridine formation [68]. 

### 3.12. Cefepime

Cefepime is a semi-synthetic, broad spectrum fourth-generation cephalosporin active against aerobic Gram-positive and Gram-negative pathogens including *P.* aeruginosa (details in Table 2). The parenteral administration of 50 mg/kg every 8 h (maximum dose: 6 g per day) is currently recommended for children and adolescents, traditionally administered as an intravenous infusion over 30 min. Neonates should receive 30 mg/kg/dose every 12 h, increasing the dose up to 50 mg/kg/dose every 12 h in case of meningitis or severe infections due to *P. aeruginosa* [69]. 

### 3.13. Pharmacokinetic/Pharmacodynamic Data 

Previous studies confirmed that in children aged 2 months to 16 years cefepime has a shorter half-life (1.26–1.93 h), compared to adults. In addition, a larger volume of distribution of cefepime was observed in the pediatric population [70]. For these reasons, strategies aimed at optimizing fT > MIC are extremely interesting. However, very few studies assessed the use of cefepime as prolonged infusion, in the pediatric population. Findings from Shoji et al. showed that only prolonged infusion (e.g., within 3 h) of 50 mg/kg guaranteed high rates of target attainment, up to 57% and 100% with every 12 and every 8 h dosing, in children older than 30 days of age and for infection due to *Enterobacteriaceae* with elevated MIC (8 mcg/mL) [71].

### 3.14. Clinical Outcomes

To date, no studies evaluated the impact of administering cefepime as prolonged infusion, among the pediatric population. Descriptive evidence published by Nichols et al., suggests that implementing extended-infusion of cefepime as standard of care is feasible and not related to major complications, in children with a median age of six years admitted to several pediatric departments, excluding intensive care units [72]. However, comparison of outcomes with patients receiving conventional intermittent infusions were beyond the scope of their study. 

No studies evaluated the tolerability and compliance of outpatients’ continuous infusion of ceftazidime with once-daily changes of infusion device. Voumard and colleagues described adult outpatients’ treatment with continuous infusion cefepime, demonstrating a good compliance profile with an effective and safe approach [38].

### 3.15. Meropenem

Meropenem is a broad-spectrum beta-lactam antibiotic of the carbapenem class active against several Gram-positive and Gram-negative aerobic and anaerobic microorganisms, including extended-spectrum betha-lactamase (ESBL) producing bacteria (details in Table 2). It has a favorable penetration profile of tissues and body fluids alike, and it is well tolerated in preterm infants, neonates and children [73,74,75]. The suggested intravenous dose for susceptible infections in infants, children and adolescents is 20 mg/kg/dose every 8 h (maximum dose 1000 mg/dose) [76]. In the case of central nervous system infections and of combined antibiotic treatment against multidrug resistant bacteria, such as carbapenemase-producer gram-negative bacteria, as much as for children affected by cystic fibrosis with pulmonary exacerbations, the pediatric recommended dose is 40 mg/kg/dose every eight hours (maximum dose 2000 mg/dose) [76,77].

The inappropriate use of meropenem has led to an increasing incidence of carbapenemase-producing *Enterobacteriaceae* (CPE), which have now become endemic worldwide, including in Europe and particularly in Italy and Greece [78]. To face with the rapid spread of CPE, several efforts including the implementation of antimicrobial stewardship programs have been strongly recommended, aimed at enhancing the selection of carbapenem-sparing regimens and, when choosing a carbapenem-based treatment, trying to optimize its use in terms of ensuring optimal serum concentrations, in order to avoid the upraise of resistant breeds [79]. 

### 3.16. Pharmacokinetic/Pharmacodynamic Data 

Meropenem is usually administered through infusion over 30 min, as some evidences indicated that this drug may undergo a degradation, few hours after reconstitution [80,81]. Nonetheless, its bactericidal activity is known to be time-dependent [82,83,84]. Evidences from paediatric studies using Monte Carlo simulation found that a four-hour infusion may be suitable for pathogens with increased MICs, such as *P. aeruginosa* [74,85]. According to the prospective, multicenter, open-label pharmacokinetics study conducted by Pettit et al., 30 children affected by cystic fibrosis and with concomitant *P. aeruginosa* infection received extended infusion (e.g., administered in 3 h) of 40 mg/kg meropenem every 8 h. At Monte Carlo simulation, meropenem administered with prolonged (3 h) infusion had a greater likelihood of obtaining 40% fT > MIC against pathogens with meropenem MICs of 1–8 mg/L, compared to intermittent (30 min) infusion [86], with a good tolerance profile.

Similar findings were reported by Cies et al. [87] that examined nine critically ill children aged one to nine years receiving meropenem: at Monte Carlo simulation only the three- to four-hour prolonged infusion and 24 h continuous infusion regimens achieved the optimal PTA (40% fT > MIC) against all susceptible Gram-negative bacteria, while increasing dosage regimens to 120–160 mg/kg/day administered as continuous infusion may be necessary to achieve a PTA of 80% fT > MIC, in critically ill children [87]. 

In their recent population pK study, Rapp et al. simulated dosing regimens of meropenem in critically ill paediatric patients with differences in renal function. Their results showed that the best regimen was continuous infusion (60 and 120 mg/kg/day), which allowed attainment of the target of 50% fT > MIC and 100% fT > MIC in patients with normal and increased creatinine clearance, with infection by germs with high MIC values (>4 mg/L), with no risks of accumulation. However, this was not demonstrated in children with a decreased creatinine clearance and severe renal failure [88]. 

### 3.17. Clinical Outcomes 

The impact of extended infusion of meropenem in terms of clinical outcomes, such as mortality or clinical improvement, among the pediatric population is still unknown, as very few studies were conducted on this field. Recent evidence seems to confirm a better clinical outcome for neonates with Gram-negative late-onset sepsis (GN-LOS) treated with prolonged infusion of meropenem, if compared to conventional intermittent dosing [89]. In fact, according to the single center, open-label RCT conducted by Shabaan et al., neonates receiving extended (4 h) infusion had a significantly higher rate of both clinical improvement and microbiologic eradication, with a significantly lower neonatal mortality and shorter duration of respiratory support, compared to conventional infusion. In addition, prolonged infusion was associated with significantly less cases of acute kidney failure (AKI) in this group [89]. Meropenem was administered at a dosing regimen of 20 mg/kg/dose every 8 h, 40 mg/kg/dose in meningitis or *Pseudomonas aeruginosa* infection. 

Pettit et al. also confirmed that meropenem administered as prolonged infusion was well tolerated, as only one patient stopped therapy due to an unspecified adverse event [86]. 

Lastly, Padari et al., in their study reported no cases of toxicity for meropenem given with continuous infusion to seriously ill premature neonates. As for mortality, there was 1/9 patient reported in the standard-infusion group and 1/10 in the prolonged-infusion group, both deaths occurred more than seven days after completion of therapy, with no relation to meropenem administration [90]. 

All studies including patient data are summarized in Table 1. Spectrum of activity, types of infections addressed, possible adverse effects of the considered antibiotics and information about the development of antibiotic resistance are detailed in Table 2. 

## 4. Conclusions

Pediatric patients represent a challenging population, showing variations in pharmacokinetics throughout the different ages. The question whether time-dependent antibiotics should be administered in continuous or intermittent infusion is still open, with few studies led on children and adult data not being fully extendable. 

Indeed, prolonged and continuous infusions of antibiotics seem to have a greater probability of target attainment as compared to intermittent infusion regimens, with generally good clinical outcomes and tolerability and safety profile, and therefore should be considered in the pediatric population on a case-to-case basis. This way of administration also has the advantage of being cost-effective on a hospital basis and may be considered to reduce the length of hospitalization in stable children, as continuous infusion treatments can be administered at home with home-based or out clinic-based once-daily changes of infusion device [19]. However, further studies are needed in order to better explore the feasibility, acceptability and impact of this drug-administration strategy, among the pediatric population. It should be taken into consideration that continuous infusion regimens may have some limitations, such as limited molecule stability for some drugs, the need for additional intravenous accesses for hospitalized patients and the association with unavoidable limited mobility of patients. 

Our narrative review has some limitations: well-designed included studies were scarce and with different designs, therefore comparison between studies was not possible. Furthermore, we only considered studies in English, leading to the possibility of missing further data published in other languages. In light of results from pK/pD studies and considering the reported safety and tolerability of prolonged or continuous antibiotic infusion, our review highlights the need of conducting randomized-controlled trials aimed at exploring the clinical impact, tolerability and patient’s acceptability of continuous administration of time-dependent antibiotic administration, compared to intermittent infusion, in the neonatal and paediatric population.

## Figures and Tables

**Table 1 antibiotics-09-00329-t001:** Characteristics of included studies, reporting data on pediatric patients on intermittent vs. continuous infusion—study design, setting, population, antibiotic dose, toxicity, outcomes.

Included Studies	Study Design	Setting	Antibiotic and Dose	Population			Primary Outcome	Toxicity	Other Secondary Outcomes
				Total (Analyzed)	Standard	Prolonged/Continuous			
*Zylbersztajn, Arch Argent Pediatr 2013* [34]	Case series	Spain, PICU	**Vancomycin** IIV 40 and 60 mg/kg/day every 8–6 h, shifted to CIV at 50 mg/kg/day	6 children 2 months–7 years	6	6	Clinical cure (all had a favourable outcome, with negativization of cultures)	No nephrotoxicity	All patients achieved levels between 10 and 25 ug/mL
*Gwee, Pediatrics 2019* [35]	RCT	Australia, NICU and PICU	**Vancomycin** IIV 15mg/kg/dose every 24, 12, 8 or 6 h or CIV 15 mg/kg loading dose followed by 20–50 mg/kg/day	111 infants	54	57	21 of 51 (41%) infants of IIV group achieved target concentrations at the first steady-state level compared with 45/53 (85%) of CIV group.	No clinically relevant adverse effects were observed in either regimen	The mean times to bacteremia clearance were 55.3 h in IIV group and 46.1 h in CIV group
*Demirel,* *Journal of Neonatal Perinatal Medicine 2015* [36]	Retrospective observational	Turkey, NICU	**Vancomycin** IIV 20mg/kg/die Or CIV 10 mg/kg loading dose followed by 20 mg/kg/day	77 preterm infants	41	36	At 48 h, 52.8% of infants of CIV group achieved vancomycin therapeutic levels, compared to 34.1% of patients in IIV group	No nephrotoxicity	No significant differences between groups, in terms of microbiological and clinical outcomes
*Hurst et al., Journal of the Pediatric Infectious Diseases Societ, 2018* [37]	Retrospective study	USA		240 children			Overall: TDD of CIV required to attain therapeutic SVCs according to age. 76/240 had a goal SVC of 10 to 15 μg/mL 164/240 had a goal of 15 to 20 μg/mL	A total of 19 patients had a 25% to 49% decrease in creatine clearance (CrCl)	
			**Vancomycin** *Goal SVC of 10–15 μg/mL:* final TDD on IIV 79.5 mg/kg/day, shifted to CIV 46.2 mg/kg/day *Goal SVC of 15*–*20 μg/mL:* final TDD on IIV 77.9 mg/kg/day, shifted to CIV 47 mg/kg/day	54 children >31 days to <2 years	54	*Goal SVC of 10–15 μg/mL:* 17 *Goal SVC of 15*–*20 μg/mL:* 37	*Goal SVC of 10–15 μg/mL:* 82% of patients achieved a therapeutic SVC *Goal SVC of 15–20 μg/mL:* *51%* of patients achieved a therapeutic SVC		Frequency of attaining goal SVCs on CIV Time to attainment of a therapeutic SVC on CIV Safety of CIV
			**Vancomycin** *Goal SVC of 10–15 μg/mL:* final TDD on IIV 79.1 mg/kg/day, shifted to CIV 44.5 mg/kg/day *Goal SVC of 15–20 μg/mL:* final TDD on IIV 78.7 mg/kg/day, shifted to CIV 45.6 mg/kg/day	94 children 2 to < 8 years	94	Goal SVC of 10–15 μg/mL: 38 Goal SVC of 15–20 μg/mL: 56	*Goal SVC of 10–15 μg/mL:* 82% of patients achieved a therapeutic SVC *Goal SVC of 15–20 μg/mL:* *41%* of patients achieved a therapeutic SVC		
			**Vancomycin** *Goal SVC of 10–15 μg/mL:* final TDD on IIV 72.5 mg/kg/day, shifted to CIV 41.5 mg/kg/day*Goal SVC of 15–20 μg/mL:* final TDD on IIV 72.9 mg/kg/day, shifted to CIV 43.1 mg/kg/day	92 children 8 to < 18 years	92	Goal SVC of 10–15 μg/mL: 21 Goal SVC of 15–20 μg/mL: 71	*Goal SVC of 10–15 μg/mL:* 67% of patients achieved a therapeutic SVC *Goal SVC of 15–20 μg/mL:* *76%* of patients achieved a therapeutic SVC	*Goal SVC of 10–15 μg/mL:* renal failure in a 10-year-old *Goal SVC of 15–20 μg/mL:* renal injury in a 17-year-old	
*Solórzano-Santos et al., Rev Invest Clin 2019* [58]	Non-blinded RCT	Mexico, third-level paediatric hospital	**Piperacillin/tazobactam**, 300 mg/kg/day IA (4 doses) versus CI of 300 mg/kg/day over 24 h (after loading dose of 75 mg/kg over 30 min)	176 episodes of febrile neutropenia in children	100(Group 1)	76 (Group 2)	Clinical cure (fever decreased in the first 48 after therapy start in 45% of patients; improvement of signs and symptoms at 72h in 80% and 73% of patients in the two groups respectively) Treatment failure (13/100 failures in Group 1 and 16/76 in Group 2)	/	No differences in fever resolution, clinical cure rate or mortality (2 patients died, one for each group).
*Knoderer et al., JPPT Clin Inv 2017* [59]	Retrospective case series	USA (general surgery, oncology)	**Piperacillin/tazobactam**, 112.5 mg/kg intravenously (IV) every 8 h, infused by EI (over 4 h)	39 children with Enterobacteriaceae related infection (mostly *E. coli* & K*lebsiella)*	/	39	Clinical cure (29/39 (74%) met clinical cure, at 21 days after TZP initiation)	/	length of stay duration of TZP treatment30-day readmission (15/39 (38.5%) had a 30 days readmission) 30-day mortality (No deaths)
*Rappaz I, Eur J Pediatr 2000* [66]	Prospective cross over study	Switzerland(Cystic Fibrosis Centre)	**Ceftazidime** Thrice-a-day 20 min 200 mg/kg/day IA for 14 days versus CI of ceftazidime 100 mg/kg/day for 14 days	14 children with cystic fibrosis (CF)	14	14	Clinical cure: efficacy of both regimens assessed by comparing surrogate markers (all patients improved clinically, no differences in terms of variation of several pulmonary, inflammatory and nutritional variable)	No clinically relevant adverse effects were observed in both regimen	Tolerability and feasibility of CI regimen; Positive impact on the quality of life of CF children
*Hubert D, Antimicrobial Agentis and Chemotherapy, 2009* [67]	multicenter, randomized crossover study	France, 15 (Cystic Fibrosis Centers)	**Ceftazidime** IA (thrice-daily) of 200 mg/kg/day versus CI, after a loading dose of 60 mg/kg	70 children with CF	34: thrice-daily ceftazidime short infusions for the first course and ceftazidime CI for the second course (group A)	36: ceftazidime CI for the first course and short infusions for the second course (group B).	Efficacy: ITT: no difference in FEV 1 (assessed at the end of therapy) between group A and group B, with +7.6% after continuous infusion and + 5.5% after short infusions) (90% CI 2.1 (−0.3 to 5.2), p 0.15) but better clinical outcome after continuous ceftazidime treatment in patients harboring resistant isolates (*p* < 0.05).	Tolerance: 124 adverse events reported (68 on SIs and 56 on CI) in 50 patients, of those only 2 were considered severe (1 after the SIs 1 after the CI)	Similar quality-of-life scores for both treatments, however 82% of the 57 patients preferred the CI administration, rather than short infusions
*Shabaan AE, Pediatr Infect Dis J 2017* [89]	Single center, open-label RCT	Egypt, NICU	**Meropenem** 60 mg/Kg/day (120 mg/kg/day if meningitis/P.aeruginosa), IA (over 30 min) in group1 versus EI (over 4 h) in group2	102 neonates (< 28 days) with late-onset sepsis due to GNB	51	51	Clinical success: 31/51 (61%) EI vs. 17/51 (33%) IA, *p* = 0.009 Odds Ratio: 3.10 (1.38, 6.96) Microbiologic success: eradication at MER 7th day (82% prolong vs. 56.8% conv, *p* = 0.009) -shorter duration of respiratory support [4 days in prolong (0–18) versus 12.5 days in conv (5.7–17.2) vs., *p* = 0.03]	Reduced risk of AKI with EI (3/51, 6%) compared to IA (12/51, 3.5%), *p* = 0.02	Mortality: 7/51 (14%) EI vs. 16/51 (31%) IA, *p* = 0.03) RR: 0.44 (0.20–0.47) *
*Padari et al., AAC, 2012* [90]	Prospective, open label study	Estonia, NICU	**Meropenem** 20 mg/kg bid over 30 min vs. 4h infusion	19 neonates (< 23w, BW < 1.2 kg)	9	10	Steady-state PK: higher C max in the short-infusion group and a higher time to drug C max in serum (T max) in the prolonged-infusion group. All other PK parameters were similar. - All of the patients in the short-infusion group and 8/10 in the long-infusion group achieved an fT MIC of 100% for an MIC of 2 mg/L. Safety of meropenem given via short or prolonged infusion	None	Mortality: 1/9 in IA vs. 1/10 in EI (> 7 days after completion of therapy) In VLBW neonates, meropenem infusions of 30 min are optimal

Abbreviations: CNS: Central Nervous System; PICU: Pediatric Intensive Care Unit; RCT: Randomized Controlled Trial; IIV: Intermittent Infusion Vancomycin; CIV: Continuous Infusion Vancomycin; SVC: Serum Vancomycin Concentration; NICU: Neonatal Intensive Care Unit; IA: Intermittent Administration; CI: Continuous Infusion; EI: Extended Infusion, TDD: Total Daily Dose; TZP: Piperacillin/Tazobactam; CF: Cystic Fibrosis.

**Table 2 antibiotics-09-00329-t002:** Spectrum of activity and infection types, adverse effects and antimicrobial resistance for the considered antibiotics.

Antibiotic	Spectrum of Activity Infection Types	Adverse Effects	Antimicrobial Resistance
**Glycopeptides**			
Vancomycin	Bactericidal for several aerobic and anaerobic gram-positive bacteria, including coagulase-negative *Staphylococcus* and *S. aureus*. Bacteriostatic for enterococci. Skin and soft tissue infections, bone and joint infections, bloodstream infections and endocarditis, CNS infections, *C. difficile* colitis (if administered orally) 10.14.	Infusion-related adverse effects: “red man syndrome”; pain in the area of injection; allergic reactions. Drug-related toxicity: neutropenia, thrombocytopenia, eosinophilia, thrombophlebitis, chills, fever, rash, nephrotoxicity, and ototoxicity [60,82,83]	VRE: selection pressure by indiscriminate use of vancomycin, linked to at least 4 genes (Van A-D); VISA/VRSA: thickened and aggregated cell walls Resistance of *S. epidermidis*: biofilm [82,83]
**β-lactams**			
Piperacillin/Tazobactam	Wide spectrum of activity against Gram-positive/Gram-negative aerobic and anaerobic pathogens, *P. aeruginosa*. Moderate to severe infections, including community and hospital acquired pneumonia, complicated pelvic and urinary tract infections, complicated skin and soft tissue infections, intra-abdominal infection, severe sepsis and septic shock. Piperacillin is largely used for the treatment of sensitive strains of *P. aeruginosa* [10,57,58,59,60].	Anaphylactic/anaphylactoid reactions; Stevens–Johnson syndrome and toxic epidermal necrolysis, drug reaction with eosinophilia; antibiotic-induced pseudomembranous colitis; bleeding, abnormalities of coagulation tests, such as clotting time, platelet aggregation and prothrombin time; leukopenia, neutropenia; nephrotoxicity [53,83].	Resistance in E. coli: inhibitor-resistant variants within the TEM and SHV β -lactamase families; overexpression of inhibitor-sensitive enzymes, such as BlaTEM-1 [82,83].
Ceftazidime	Gram-negative germs, such as *P. aeruginosa*, *E. cloacae*, *E. coli*, *H. influenzae, K. pneumoniae*. Febrile neutropenia in children, respiratory tract infections, especially in children with CF that present chronic infections by P. aeruginosa [60,61,62,63].	Maculopapular or morbilliform skin eruptions, drug fever, and a positive Coombs test; anaphylaxis; granulocytopenia; renal toxicity; disulfiram-like reactions [83].	Inactivation by bacterial β-lactamases, alteration of PBPs, and alteration of bacterial permeability to cephalosporins: susceptible to hydrolysis by the inducible, chromosomally encoded β-lactamases and the plasmid extended-spectrum β-lactamases [83].
Cefepime	Enterobacteriaceae, *E. coli*, *Proteus* spp, *Klebsiella* spp, *S. pneumoniae*, MSSA strains, multidrug-resistant Gram-negative bacteria, such as AmpC β -lactamase-producing strains and several strains of ESBL-producing organisms. Urinary tract and lower respiratory infections in children [66,70].	Maculopapular or morbilliform skin eruptions, drug fever, and a positive Coombs test; anaphylaxis; granulocytopenia; renal toxicity; disulfiram-like reactions [83].	inactivation by bacterial β-lactamases, alteration of PBPs, and alteration of bacterial permeability to cephalosporins; poor inducer of type I β-lactamases and less susceptible to hydrolysis [83].
Meropenem	Gram-negative and Gram-positive microorganisms Enterobacteriaceae, *P.aeruginosa*, *Bacteroides* spp, *H. influenzae*, *N, gonorrheae, S. aureus, s. epidermidis, S.saprophiticus,* coagulase negative streptococci. In combination with other antibacterial agents: MSSA, *S. pyogenes, S. agalactiae, S, pneumonia. E. faecium* strains are resistant. Meningitis, intra-abdominal infections, lower respiratory infections, bacteremia and sepsis [75].	Seizures, CNS adverse events, diarrhea, rash, nausea, and vomiting [83].	Hydrolyzed by the β-lactamases of S. maltophilia. Does not bind to the PBPs of *E. faecium*. Resistance due to decreased permeability is uncommon [83].

Abbreviations: CNS: Central Nervous System; MRSA: Methicillin Resistant *S. aureus*; VRE: Vancomycin-resistant Enterococci; VISA/VRSA: Vancomycin Intermediate/Resistant *S. aureus*; PBP: Penicillin Binding Proteins; CF: Cystic Fibrosis; MSSA: Methicillin Sensitive *S. aureus*; ESBL: Extended-Spectrum β-Lactamase.

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
