# Peer review of "Optimizing Antibiotic Treatment Strategies for Neonates and Children: Does Implementing Extended or Prolonged Infusion Provide any Advantage?"

_antibiotics, 2020, doi:10.3390/antibiotics9060329_

Round 1

Reviewer 1 Report

Costenaro et al. summarized recent new findings regarding the optimization of the antibiotic treatment strategies for neonates and children, comparing the advantages of extended or prolonged infusion treatments.

The manuscript is well-written and organized, however more information should be provided in order to improve the manuscript, in particular:

  • Please, indicate the type of infections that could be treated, in the pediatric population, with the antimicrobials cited in the manuscript specifying the bacterial species which are involved in that infections. For example,
  • 8 lines 194 196) which kind of severe infections? The authors wrote Gram- positive and Gram-negative bacteria, could you report some examples based on the data present in literature?
  • 10 lines 266 “pathogen isolated in sputum” which pathogens? Maybe P. aeruginosa or S. maltophilia?
  • Please add more information and comments about the possible side effects (reported in table 1) of the antimicrobials used in the pediatric population as well as more information about the possible development of antimicrobial resistance.
  • Please add a more descriptive legend to table 1.
  • Please, check the References accurately, according to the journal guidelines.

Author Response

According to Reviewer 1’s comments:

  • We indicated the spectrum of activity, specifying the bacterial species involved, and types of infections that could be treated in the pediatric population in Table 2 (Line 106), see below (we reported the table for exact changes at the bottom of this letter): Lines 100-101, “Spectrum of activity, types of infections addressed, possible adverse effects of the considered antibiotics and information about development of antibiotic resistance are detailed in Table 2. ”
  • Vancomycin: Table 2 (Line 106); in text: “Details about spectrum of activity, adverse effects and hints to antibiotic resistance can be found in Table 2.” (Lines 115-116).
  • Piperacillin/Tazobactam: Table 2 (Line 106), in text: “(details in Table 2)”, Line 196.
  • Ceftazidime: Table 2 (Line 106); In text: “Details can be found in Table 2”, Line 235; “ aeruginosa”, Line 268.
  • Cefepime: Table 2 (Line 106); in text: “Details in Table 2” Line 282; modified references: Lines 286, 291, 296.
  • Meropenem: Table 2 (Line 106); “(details in Table 2)”Line 312

  • We added information about the possible side effects and more information about the possible development of antimicrobial resistance, reporting them in the same table (Table 2, Line 106), also addressing one of the comments of Reviewer 2, that suggested presenting more data in tables because of the heavy structure of the manuscript.

  • We added a more descriptive legend to Table 1: Line 103: We extended, specifying “Characteristics of included studies, reporting data on pediatric patients on intermittent vs continuous infusion – Study design, setting, population, antibiotic dose, toxicity, outcomes”

  • We revised bibliography according to the Instructions for Authors (Track changes in manuscript from line 417 to line 593).

Please see the attachment for table 1 and table 2

Reviewer 2 Report

The review paper submitted by Daniele Donà and coauthors deals with with prospective and retrospective studies on infusion strategies of various antibiotics in the children population. The manuscript is quite well written but must be improved before the publication. My major concern is that the authors selected only nine articles. Please explain if that limited number of research groups there will not affect the actual image of the problem. Narrowing the selection of studies only to those described in English may also lead to wrong conclusions. In my opinion, authors should take the effort to translate some significant works in other languages as well and include them in the analysis. Moreover, the form in which it is written does not attract the attention of readers and will not ensure a large number of citations in the future. I suggest presenting more data in the tables (there is only one table in the current form of the manuscript) and summarizing and illustrating problems with some figures.

Author Response

As for Reviewer 2’s comments:

  • We remade the search in order to include published articles comparing intermittent vs continuous infusion of the considered antibiotics in children, in English as well as languages other than English. We only found one relevant article in Spanish, a case series on 6 patients treated with intermittent and then continuous infusion vancomycin, to be included in our narrative review. We found no other eligible articles for translation. Published studies on pediatric population on the topic are indeed few, and we believe that results may not actually affect the real image of the problem.

 In text:

Line 83: we omitted from search methods “and “languages: English””;

Line 92: we omitted from selection criteria “Furthermore, publications of language different from English language were not considered. ”

Line 98-99: “Of 38,906 titles and abstracts, 114 were eligible for inclusion in this review, and ten studies were included, nine published in English and one in Spanish.”

Since Table 1 was not editable in text, we uploaded the updated version (see below), in text: from Line 103.

  • In order to make the text lighter, we presented and expanded details about spectrum of activity of the mentioned antibiotics, as wells as information about treated infections, in Table 2, Line 106, which we also completed following Reviewer 1’s comments.

Please see the attachment for tables 1 and 2.

Round 2

Reviewer 2 Report

The authors have responded to my comments and have made appropriate corrections. The manuscript deals with very important issues, so I think it should be published in its current form.

Author Response

On behalf of all authors, I thank you for your endorsement.

Dr Paola Costenaro